# Magnetic Resonance Imaging (MRI)-Based Semi-Quantitative Methods for Rheumatoid Arthritis: From Scoring to Measurement

**DOI:** 10.3390/jcm13144137

**Published:** 2024-07-16

**Authors:** Fausto Salaffi, Marina Carotti, Marco Di Carlo, Luca Ceccarelli, Sonia Farah, Anna Claudia Poliseno, Andrea Di Matteo, Francesca Bandinelli, Andrea Giovagnoni

**Affiliations:** 1Rheumatology Unit, Department of Clinical and Molecular Sciences, Università Politecnica delle Marche, Carlo Urbani Hospital, Jesi, 60035 Ancona, Italy; fausto.salaffi@gmail.com (F.S.); sonia.farah91@gmail.com (S.F.); 2Department of Clinical, Special and Dental Sciences, Università Politecnica delle Marche, 60126 Ancona, Italy; marina.carotti@gmail.com (M.C.); annaclaudia.poliseno@gmail.com (A.C.P.); a.giovagnoni@univpm.it (A.G.); 3Division of Radiology, Department of Radiological Sciences, University Hospital Azienda Ospedaliera Universitaria delle Marche, 60126 Ancona, Italy; 4Oncohematologic and Emergency Radiology Unit, Department of Pediatric and Adult Cardio-Thoracic and Vascular, IRCCS Policlinico di Sant’Orsola, 40138 Bologna, Italy; luca.ceccarelli28@gmail.com; 5Leeds Institute of Rheumatic and Musculoskeletal Medicine, University of Leeds, Leeds LS2 9JT, UK; andrea.dimatteo@hotmail.com; 6Leeds Biomedical Research Centre, National Institute for Health Research, Leeds Teaching Hospitals NHS Trust, Leeds LS9 7TF, UK; 7Rheumatology Department, USL Tuscany Center, San Giovanni di Dio Hospital, 50143 Florence, Italy; francesca.bandi@gmail.com

**Keywords:** rheumatoid arthritis, magnetic resonance imaging, synovitis, erosion

## Abstract

Rheumatoid arthritis (RA) is a chronic autoimmune disease that primarily affects the small joints of the hands and feet, characterized by pain, inflammation, and joint damage. In this context, magnetic resonance imaging (MRI) is useful to identify and monitor joint/tendon inflammation and the evolution of joint damage, playing a key role in treatment response evaluation, in addition to clinical measurements. Various methods to quantify joint inflammation and damage with MRI in RA have been developed, such as RA-MRI Score (RAMRIS), Early RA-MRI Score (ERAMRS), and Simplified RA-MRI Score (SAMIS). RAMRIS, introduced in 2002, offers an objective means to assess inflammation and damage via MRI in RA trials, encompassing findings such as synovitis, bone erosion, and edema/osteitis. Recently, an updated RAMRIS version was developed, which also includes the evaluation of joint space narrowing and tenosynovitis. The RAMRIS-5, which is a condensed RAMSIS version focusing on five hand joints only, has been proven to be a valuable resource for the semi-quantitative evaluation of RA joint damage, both in early and established disease. This narrative literature review will provide an overview of the MRI scoring systems that have been developed for the assessment of joint inflammation and structural damage in RA patients.

## 1. Introduction

Rheumatoid arthritis (RA) is a chronic autoimmune condition primarily affecting the small joints of the hands and feet, leading to pain, inflammation, and joint deterioration and resulting in diminished quality of life [1,2,3,4]. In scientific research involving RA patients, magnetic resonance imaging (MRI) is gaining prominence due to its sensitivity in detecting inflammation and tissue damage [5].

While conventional radiography remains the standard in everyday practice for RA evaluation [6], it primarily identifies late-stage disease activity and structural damage, prompting the adoption of additional medical imaging methods like MRI and ultrasonography (US) to detect early symptoms [7,8,9]. Radiographs are inadequate for detecting early signs of joint inflammation like synovitis, bone marrow edema (BME), or pre-erosion. On the other hand, MRI is highly effective in identifying these changes. Notably, even in clinical remission, MRI can reveal signs of disease activity with significant prognostic implications [10,11], firmly establishing MRI as a crucial tool in RA diagnosis [12,13].

The RA-MRI-Scoring (RAMRIS) system has greatly facilitated the use of MRI in RA outcome studies. It provides a semi-quantitative standardized assessment of inflammatory soft tissue changes and bone destruction [14,15,16]. RAMRIS evaluates 23 joint sites in the hand, providing detailed sub-scores for synovitis, bone marrow edema (BME), erosions, and more recently, tenosynovitis and joint space narrowing (JSN). Each condition is graded on a scale, with synovitis and BME scored from 0 to 3 and erosions from 0 to 10 [17]. Despite its comprehensiveness, RAMRIS is time-consuming and demands experience for reproducibility. To address these issues, RAMRIS-5 and the Simplified Rheumatoid Arthritis Magnetic Resonance Imaging Score (SAMIS) were introduced [18]. In patients with established RA who have been suffering from the disease for five years or more, low-field MRI using the RAMRIS-5 offers a time-efficient alternative that closely correlates with the standard RAMRIS [19]. The RAMRIS-5 targets five specific joint sites to assess bone edema and erosion—namely, the third metacarpophalangeal (MCP) joint (1), the second MCP joint (2), the capitate bone (3), the triquetral bone (4), and the distal ulna. It also includes scoring for synovitis at the second and third MCP joints and the wrist.

SAMIS streamlines scoring, evaluating one hand and using a 1 to 10 scale for grading erosions. Edema and synovitis scores range from 0 to 1 and from 0 to 2, respectively. Additionally, the Early Rheumatoid Arthritis Magnetic Resonance Imaging Score (ERAMRS) was developed to assess wrist MRI inflammation in early RA (ERA), covering 15 wrist bones, 7 wrist joints, and 9 wrist tendons. Compared to existing MR scoring systems, ERAMRS offers greater efficiency, reliability, and better correlation with clinical scoring systems and serological markers of inflammation [20].

This literature narrative review provides a historical overview of prominent scoring techniques and explores the utility of MRI in diagnosing and detecting early structural changes in RA. It aims to equip radiologists with the skills to produce concise reports and communicate results effectively.

## 2. Materials and Methods

The articles selected for this narrative literature review were searched for on PubMed in December 2023 using the query [“rheumatoid arthritis” AND “magnetic resonance imaging” AND “scoring systems”]. Subsequently, original studies pertaining to the semi-quantitative scoring systems—RAMRIS, RAMRIS-5, SAMIS, and ERAMRS—were selected. Due to the non-systematic nature of the review, we will not provide the flow chart of the selection of the literature, even if the 2020 PRISMA guidelines were followed for eligibility criteria.

## 3. Results

MRI is a pivotal imaging modality for assessing RA disease activity. MRI offers comprehensive evaluation of all joints, standardized assessments, and quantifiable data on inflammation. However, it comes with certain drawbacks, including high costs, the need for intravenous contrast agents, and extended diagnostic procedures. Moreover, RA patients might find whole-body, high-field MRI scans less accommodating, especially when repeated examinations are necessary. In contrast, extremity-specific MRI equipment proves cost-effective, patient-friendly, and ideal for monitoring treatment progress. Recent advancements in whole-body MRI have enriched our understanding of RA pathogenesis and treatment outcomes [21].

The hand and wrist serve as pivotal anatomical regions for RA assessment and have been the primary focus of numerous studies [22,23,24,25,26]. These joints hold particular significance as they are frequently affected in the early stages of the disease, are involved in nearly all RA cases, and offer valuable insights into overall RA-related inflammation [27,28,29,30].

The RAMRIS, developed by OMERACT for assessing RA, includes evaluations for synovitis, BME, and erosions. Synovitis is characterized by an enhanced synovial compartment that exceeds the width of normal synovium after the infusion of a contrast agent. This condition is evaluated in each MCP joint and three specific wrist regions: the distal radioulnar joint, the radiocarpal joint, and the combined intercarpal and carpometacarpal joints. BME is identified as a lesion within the trabecular bone that has unclear margins and shows a high-intensity water content signal. Its severity is rated on a scale from 0 to 3, depending on the extent of bone involvement. Erosions are depicted as well-defined bone defects that are visible on T1-weighted MRI images in two different planes, often showing cortical breaks. The severity of erosions is scored on a scale from 0 to 10, which corresponds to the percentage of the bone volume affected (for instance, 10%, 20%, and so forth). Notably, a sub-score for tenosynovitis evaluation was recently introduced [17]. The cumulative score offers a comprehensive assessment of RA, encompassing both disease activity and damage. However, a drawback of RAMRIS is its time-consuming nature and potential for low reproducibility. Furthermore, RAMRIS is currently validated only for wrist and MCP joints and not yet for metatarsophalangeal (MTP) joints [31,32,33], despite the prevalence of MTP-joint inflammation [34,35].

Studies involving foot MRI in RA have yielded intriguing results [36], particularly in assessing the predictive value of MRI lesions for radiological damage, as demonstrated by Mundwiler et al. [37]. Ostendorf et al. [35] have extended the application of the RAMRIS system to the feet, affirming the highly acceptable reliability of inter-reader and intra-reader agreement in the assessment of the rheumatoid foot using the RAMRIS method.

### 3.1. RAMRIS

To assess the efficacy of treatment, the OMERACT group introduced the RAMRIS, a reliable, standardized, and semi-quantitative tool [14,17,38] (Table 1). RAMRIS evaluates 23 joint sites in the hand, including MCP joints 2–5, carpo-metacarpophalangeal joints 1–5, radiocarpal joints, intercarpal joints, and the radioulnar joint. The scoring system offers individual sub-scores for each joint in three categories: synovitis, graded from 0 to 3; BME, also graded from 0 to 3; and erosions, with a grading scale from 0 to 10 [14].

Bone erosions are characterized by sharp margins, visible in two planes (if available), and at least one cortical break. They are graded on a scale of 0 to 10 based on the erosion’s volume as a percentage of the “assessed bone volume,” measured from the articular surface cortex to a depth of 1 cm for long bones [39]. Bone edema, a lesion with ill-defined margins and high signal intensity on T2-weighted fat-saturated or short tau inversion recovery (STIR) sequences, is also scored individually, ranging from 0 to 3 based on the extent of edematous bone (0 represents no edema; 1 represents 1 to 33 percent of edematous bone; 2 represents 34 to 66 percent of edematous bone; and 3 represents 67 to 100 percent). Every bone is given a unique score (as for erosions).

Synovitis in the synovial compartment is assessed based on the gadolinium enhancement thickness exceeding the joint capsule width, graded 0 to 3 (normal, mild, moderate, severe). The OMERACT MRI in the Arthritis Working Group has revised the RAMRIS scoring systems and definitions to include updates for RA pathologies such as tenosynovitis and JSN (Figure 1) [17].

The updated RAMRIS demonstrates improved reliability and utility in alignment with the OMERACT filter [40]. It incorporates MRI acquisition enhancements and associations with patient-reported measures like pain and functional ability. New definitions and scoring techniques for additional pathologies, such as tenosynovitis, have been introduced [12,17,41].

For the forefeet MRI, where inflammatory and morphological changes’ relationship with disease activity and response to DMARD therapy is less explored, the established RAMRIS method for the clinically dominant hand is typically used. However, MRI has revealed that foot joint inflammation is as equally prevalent as in the rheumatoid hand, even in the absence of inflammatory MRI findings in the most clinically affected hand or during remission according to the Disease Activity Score—28 joints (DAS28) [42].

The region of the synovial compartment that exhibits enhancement following gadolinium that is thicker than the joint capsule’s breadth is called synovitis. Synovitis global scores are assessed in the tibiotalar joint, subtalar joint, talonavicular joint, calcaneocuboid joint, tarsometatarsal joint, cuneonavicular joint, and each MTP joint. The erosions and edema of all MTP joints and hindfoot joints are evaluated, with proximal and distal portions of the MTP joints scored separately. Tarsal bones, including the navicular, cuboid, the three cuneiforms, talus, and calcaneus, have their bases scored for erosion and edema. RAMRIS status and change scores are used to assess BME, synovitis, tenosynovitis, and erosions of the MTP joints [43,44], showing promise for its application in MTP-joint trials in early RA. However, RAMRIS evaluations can be time- and resource-intensive.

### 3.2. RAMRIS-5

In patients with established RA, with a minimum disease duration of five years and undergoing low-field MRI, Schleich et al. [19] determined that RAMRIS-5 is a time-efficient alternative that aligns closely with the standard RAMRIS. The RAMRIS-5 specifically evaluates the most clinically involved wrist and hand joints, targeting MCP 2 and 3 to assess synovitis, erosions, and BME (Figure 2). It also examines the distal ulna, the capitate bone, and the triquetral bone for erosions and BME, while the intercarpal and radiocarpal joints are considered a single site to assess synovitis.

Frenken et al. [45] demonstrated a significant correlation between the total mean scores of RAMRIS and RAMRIS-5 at baseline, 3 months, and 6 months after initiating MTX therapy. This illustrates that RAMRIS-5 is a suitable and time-saving alternative to RAMRIS for both early and established RA patients. It effectively identifies disease-typical features and facilitates follow-up assessments during treatment. RAMRIS-5 performs comparably to RAMRIS in identifying therapy-induced changes, even after three months, across all subgroups (BME, synovitis, and erosion). The differences between RAMRIS and RAMRIS-5 are minimal, particularly in edema and erosion, even after six months. The most noticeable distinction occurs in synovitis assessment during the extended 6-month follow-up. RAMRIS-5 exhibits more pronounced changes in synovitis compared to RAMRIS, resulting in a greater reduction in RAMRIS-5 scores post-therapy.

### 3.3. SAMIS

SAMIS was realized to simplify the MRI scoring process while ensuring strong intra- and inter-reader reliability, on par with the OMERACT RA-MRI scoring system. SAMIS reduces the number of assessed areas, focusing on evaluating one hand and utilizing the radiographic Simple Erosion Narrowing Score (SENS) [18,46,47]. SAMIS provides a streamlined approach for assessing RA in the hand and wrist. It involves examining specific areas including the metacarpal heads and phalangeal bases of the second through fifth MCP joints; the base of the first metacarpal; key carpal bones like the trapezium, scaphoid, and lunate; and the distal ends of the ulna and radius. For synovitis, assessments cover the intracarpal, radiocarpal, and distal radioulnar joints, along with the second to fifth MCP joints and the combined carpal joints. The scoring for erosion ranges from 1 to 10, reflecting the extent of bone damage, with a focus on juxta-articular bone lesions, sharp margins, and cortical breaks visible in MRI images. Furthermore, the SAMIS methodology includes scoring for BME and synovitis. BME is scored from 0 to 1 based on the presence of increased water content signal on T2-weighted fat-saturated or STIR images, while synovitis is assessed more thoroughly with scores ranging from 0 to 3, based on the extent of post-gadolinium (Gd) enhancement compared to normal synovium.

To reduce imaging time, invasiveness, and costs, some studies explored the accuracy of assessing RA joint pathologies using unenhanced MRI images instead of Gd-enhanced MRI (considered the reference method). Gd is typically recommended for assessing RA joint changes. It has been found that the administration of Gd contrast for MRI reduced the reliability of synovitis scores but had no impact on erosion and edema scores [42,48,49]. This limitation led to the assessment of the presence or absence of synovitis without grading in the modified SAMIS scoring system [50]. The use of Gd in MRI scans, although beneficial for detailed imaging, can significantly extend the examination time, escalate costs, increase the invasiveness of the procedure, and cause discomfort for patients. These factors collectively diminish the practicality of MRI in the routine management of RA. In a modified version of SAMIS, to save time, only the hand that was more painful or the dominant hand was evaluated [50]. MRI proved considerably more effective than conventional radiography in tracking joint damage progression in RA, whether assessing unilateral or bilateral wrist and MCP joints or unilateral MTP joints alone. Erosions in the SAMIS scoring system are quantified based on the proportion of the eroded bone relative to the total bone volume, using a scale from 0 to 3: (0) indicates no erosion; (1) signifies 11 to 33% of the bone is eroded; (2) represents 33 to 66% of the bone is eroded; and (3) denotes more than 66% of the bone is eroded. Additionally, BME and synovitis are evaluated on scales from 0 to 2 and from 0 to 1, respectively. The system consists of three sub-scores: SAMIS synovitis, SAMIS-ERO, and SAMIS-BME. Notably, these assessments are conducted without the use of contrast injection (Figure 3). The proposed modified SAMIS demonstrated excellent inter-reader reliability.

### 3.4. ERAMRS

The ERAMRS was developed to assess its clinical relevance and correlation with other MR scoring systems [20]. ERAMRS incorporated new features alongside elements from relevant existing systems specific to Early Rheumatoid Arthritis (ERA).

Synovitis evaluation: on post-contrast T1-weighted axial fat-suppressed images, synovial proliferation and enhancement were assessed in six joint areas in each wrist. These areas included the distal radioulnar joint, radiocarpal joint, intercarpal joint, first and second carpometacarpal joints, and the piso-triquetral joint. The degree of synovial proliferation was scored as 0, 1, 2, or 3, based on whether it was absent, mild, moderate, or severe, considering the expected maximum synovial proliferation for that joint area.

Synovial enhancement: similar to synovial proliferation, each of the six joint areas was assigned a score for synovial enhancement based on the degree of enhancement compared to the anticipated maximum enhancement. Scores ranged from 0 to 3.

Tenosynovial proliferation: the tenosynovial proliferation of the six extensor tendon compartments and three flexor tendon areas was scored. These included the extensor tendon compartments 1 through 6 and three flexor tendon areas: flexor pollicis longus tendon, flexor digitorum profundus, and flexor digitorum superficialis. Scores of 0, 1, 2, or 3 were assigned based on the degree of tenosynovial proliferation, with thickness measurements from the tendon’s outer border to the enhancing tendon sheath’s outer border.

Tenosynovial enhancement: each tendon group’s tenosynovial enhancement was scored similarly to synovial enhancement.

BME was assessed in 15 bone areas using T2-weighted fat-suppressed coronal images. These areas included the distal 1 cm of the radius and ulna, all eight carpal bones, and the proximal 1 cm of the five metacarpal bones. Scores ranged from B0 (no edema) to B1 (edema affecting less than 50% of the bone area) to B2 (edema affecting more than 50% of the bone area).

Scoring components and maximum score: (1) the synovial proliferation had a max score of 18; (2) the synovial enhancement had a max score of 18; the tenosynovial proliferation had a max score of 27; the tenosynovial enhancement had a max score of 27; and the BME had a max score of 30.

The maximum ERAMRS was 120 overall, with components including BME, synovial proliferation, synovial enhancement, tenosynovial proliferation, and tenosynovial enhancement. An ERAMRS test typically took around 5 min to complete.

ERAMRS provided a comprehensive assessment of joint pathologies in ERA, contributing to a better understanding of disease progression and response to treatment. This scoring system offered a quicker, more reliable, and clinically relevant alternative to other MRI scoring systems, showing strong correlations with clinical scoring systems and serological markers of inflammation.

## 4. Conclusions

MRI has established itself as a crucial diagnostic tool in RA, providing superior sensitivity for detecting key indicators of the disease such as bone erosion, BME, synovitis, and tenosynovitis. Unlike traditional radiography, MRI can identify early joint damage and delve into the underlying inflammatory processes like osteitis and synovitis that drive bone erosion and cartilage loss. This makes MRI not only highly sensitive but also extremely practical for monitoring disease progression and evaluating treatment responses, capturing subtle changes that might escape detection through standard radiographic techniques.

Furthermore, MRI facilitates a comprehensive assessment of RA, encompassing both inflammation and structural damage, which offers a more complete picture of the disease dynamics. This comprehensive view is crucial in differentiating the effects of drugs on reducing inflammation versus actual structural damage, aiding significantly in refining treatment strategies, and assessing the efficacy of therapeutic interventions. The use of the RAMRIS developed by OMERACT enhances this by providing a standardized and semi-quantitative grading system, although it can be complex and time-consuming for routine use. To overcome these challenges, simplified versions like RAMRIS-5 and SAMIS have been introduced, which streamline the scoring process while maintaining accuracy. Continual efforts towards standardizing MRI procedures and enhancing image quality are essential to maximize the utility and precision of these semi-quantitative methods in both clinical trials and everyday clinical practice.

## 5. Future Directions

Finally, some new perspectives on MRI innovative applications might out-light fascinating challenges for future definitions of semi-quantitative measurements and scores.

Firstly, whole-body magnetic resonance imaging (WB-MRI) for small body segments, consisting of a dedicated reconstruction software of a series of high-spatial-resolution images with a field of view from 25 to 50 cm [51], was recently considered the gold-standard technique for the study of pediatric juvenile idiopathic arthritis at the onset, due to its high-grade of sensibility and lack of radiations exposure risk [52,53].

In fact, the rapidity of WB-MRI sequences, reducing the amount of time necessary for traditional machines, had led to its possible indication in other musculoskeletal diseases [51], in case of the necessity to evaluate subjects with poor compliance—as, for instance, the elder population—and in multifocal inflammatory involvement.

Furthermore, the optimization of the MRI measurement of cartilage thickness, with the traction technique [54] and a three-dimensional quantitative machine [55], has recently allowed the extension of its use from osteoarthritis to systemic sclerosis patients with articular involvement [55].

Even though this initial evidence did not show significant differences from the mechanical disease [55], in the future, the hypothetical employment of cartilage thickness scores might be studied in early undifferentiated arthritis, evolving erosive RA.

Finally, the future optimization of the recent innovative employment of MRI contrast [55], as a diffusion tensor and in dynamic contrast-enhanced sequences [56,57], might improve the quantification of synovial inflammation, thereby ameliorating the performance of actual scores, with a selective indication in the early phase of RA or in false-negative remittent patients in the case of the choice of drug decalage or switching.

## Figures and Tables

**Figure 1 jcm-13-04137-f001:**
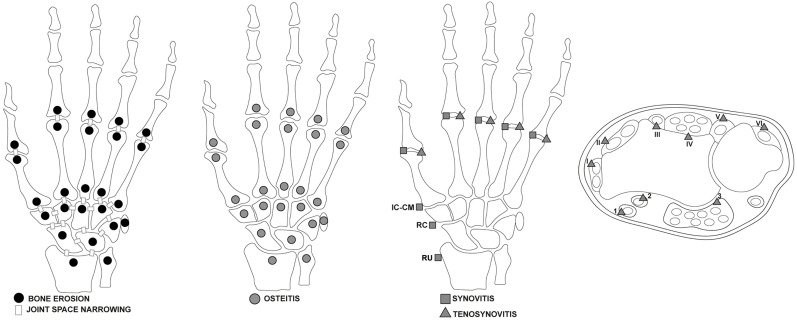
Areas of wrists and hands affected by RA illustrated by the 2016 updated Rheumatoid Arthritis Magnetic Resonance Imaging Score (RAMRIS) [17], assessed for bone erosion, joint space narrowing, osteitis, synovitis, and tenosynovitis, including extensor and flexor tendon compartments. Flexor Tendon Areas: 1: Flexor carpi radialis. 2: Flexor pollicis longus (tendon) within the radial bursa. 3: Ulnar bursa, which includes the tendon quartets of flexor digitorum profundus and superficialis. Extensor Tendon Compartments: I: Includes extensor pollicis brevis and abductor pollicis longus. II: Comprises extensor carpi radialis brevis and extensor carpi radialis longus. III: Extensor pollicis longus. IV: Extensor digitorum communis and extensor indicus proprius. V: Extensor digiti quinti proprius. VI: Extensor carpi ulnaris. Legend: RU = radio-ulnar joint; RC = radio-carpal joints; IC-CM = inter-carpal and carpo-metacarpal joints.

**Figure 2 jcm-13-04137-f002:**
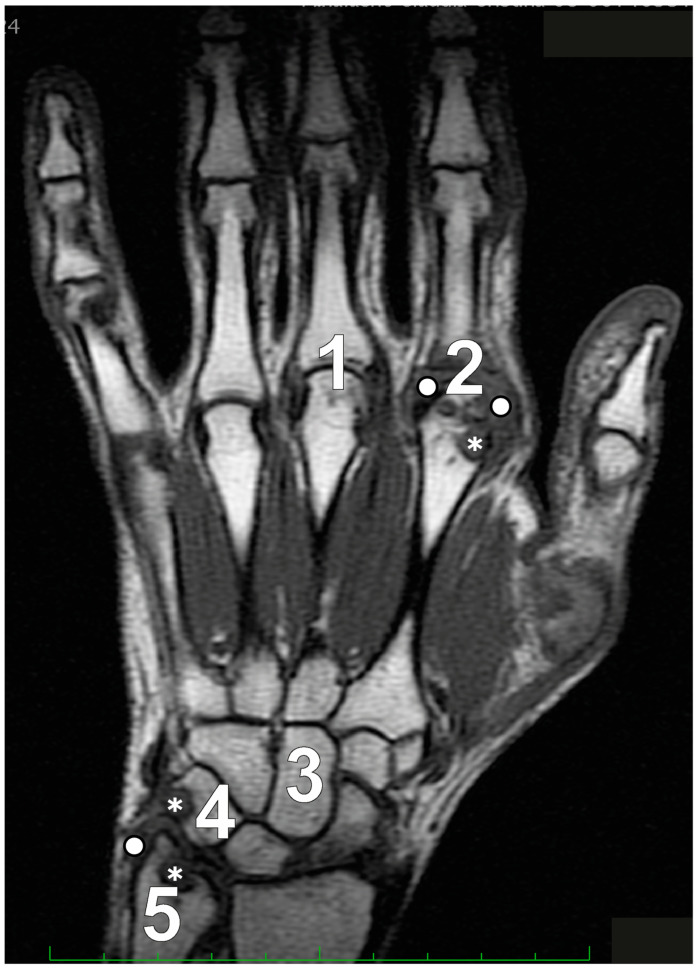
The Rheumatoid Arthritis Magnetic Resonance Imaging Score (RAMRIS)-5 specifically assesses five key sites for bone marrow edema and erosion: the third metacarpophalangeal joint (MCP III) (1), the second metacarpophalangeal joint (MCP II) (2), the capitate bone (3), the triquetral bone (4), and the distal ulna (5). Additionally, synovitis is evaluated in the second and third MCP joints along with the wrist. Legend: asterisks depict erosive changes and white circles depict synovitis.

**Figure 3 jcm-13-04137-f003:**
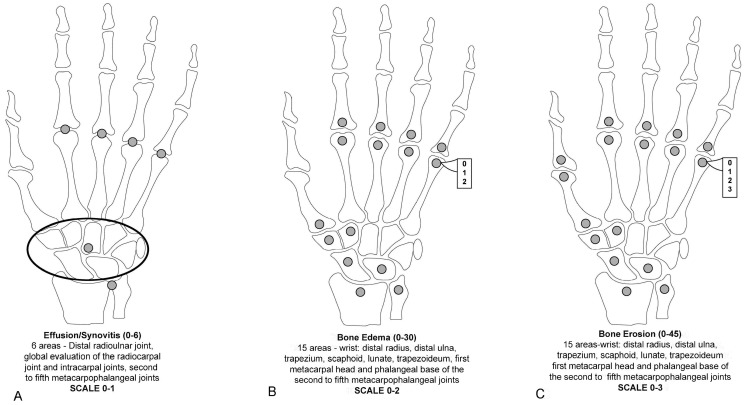
The modified Simplified Rheumatoid Arthritis Magnetic Resonance Imaging Score (SAMIS) without contrast includes the presence or absence of synovitis (**A**), the bone marrow edema semiquantitative evaluation (**B**), and bone erosion (**C**).

**Table 1 jcm-13-04137-t001:** The OMERACT RA-MRI group’s 2016 “core set” of essential MRI sequences along with definitions used in the RA-MRI scoring system (OMERACT 2016 RAMRIS) [17].

Feature	Description	Scoring (RAMRIS Units)
Synovitis	Soft tissue, characterized by increased thickness or volume, visible on T1-weighted images, and with an elevated water content—which appears as a high signal on fat-suppressed T2-weighted images—is assessed for inflammation. Three regions of the wrist are evaluated: the radio-carpal, the intercarpal–metacarpal, and the distal radio-ulnar joints of either the dominant wrist or the most inflamed wrist. For the hand, an evaluation is conducted on the MCP joints 2–5 of either the dominant side or the side that is most inflamed. This evaluation can be performed with or without gadolinium enhancement, which shows a signal intensity increase at 4–5 min post-injection.	0 (normal) to 3 (mild, moderate, severe) for each region/joint; maximum: 21
Osteitis	Recognized within the subchondral trabecular bone, this lesion presents with poorly defined margins and characteristics of a signal suggestive of increased water content, which may also appear in association with erosion. The lesion appears as a high signal on fat-suppressed T2-weighted and STIR MRI images and as a low signal on T1-weighted images. Each bone in either the dominant or the most inflamed hand–wrist is scored separately to assess these features.	0 (normal), (1) 1–33% of bone, (2) 34–66% of bone, and (3) 67–100% of bone showing increased water content. Maximum: 69 (45 for wrist alone)
Erosions	Sharply defined bone lesions located adjacent to joints, visible in two different imaging planes, are characterized by a visible cortical break in at least one plane. These lesions exhibit a reduction in the normal low signal intensity typically seen in cortical bone on T1-weighted MRI scans, as well as a decrease in the high signal observed on T2-weighted scans. Similar to the evaluation method used for osteitis, each bone in the wrist and hand is individually assessed to identify and score these specific features.	0–10, according to 10% increments of bone eroded. Maximum: 230 (150 for wrist alone)
Tenosynovitis	Tenosynovitis assessment covers six extensor tendon and three flexor tendon compartments, ranging from the radioulnar joint to the hook of the hamate (wrist). For the MCP joints, the evaluation of flexor tendons is conducted within a region that extends from 1 cm proximal to 1 cm distal to each joint. The scoring of tenosynovitis is determined by measuring the maximum width of the effusion and/or tenosynovial enhancement, with measurements taken perpendicular to the tendon’s orientation.	0–3 scale: 0 indicating no presence; 1 representing a peritendinous effusion and/or postcontrast tenosynovial enhancement of less than 1.5 mm; 2 for effusion or enhancement that is equal to or greater than 1.5 mm but less than 3 mm; and 3 for effusion or enhancement that is 3 mm or greater

Abbreviations: OMERACT = Outcome Measures in Rheumatology; MRI = magnetic resonance imaging; RAMRIS = Rheumatoid Arthritis Magnetic Resonance Imaging Score; MCP = metacarpophalangeal; Gd = gadolinium; STIR = short tau inversion recovery.

## Data Availability

The data are available if requested.

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
