# Peer review of "Magnetic Resonance Imaging (MRI)-Based Semi-Quantitative Methods for Rheumatoid Arthritis: From Scoring to Measurement"

_jcm, 2024, doi:10.3390/jcm13144137_

Round 1

Reviewer 1 Report

Comments and Suggestions for Authors

Dear authors,

Thank you for the opportunity to review the manuscript entitled “Magnetic Resonance Imaging (MRI)-Based Semi-Quantitative Methods for Rheumatoid Arthritis: From Scoring to Measurement. “

From a structural point of view, the present work falls into the category of narrative review. The manuscript is easy to read, the scientific information being presented in a logical sequence.

All the chapters of the manuscript are exhaustive and provides important data regarding the debated topic. The Introduction is well documented and provides the necessary background to accommodate the reader with the subject of the study. The purpose of the article is presented clearly and concisely. The Materials and Methods must be reviewed and some additional data related to the included items must be entered. As a radiologist, I particularly appreciate the Results/Discussions chapter, which stands out for the way it is presented and for the details provided. I think that this work is useful for both radiologists and rheumatologists. The conclusions are well formulated and reflect the results of the study.  

In addition, I identified several aspects that can improve the quality of this article:

*         Lines 82-85 - contain the same information as lines 77-80. I recommend merging them in the Introduction chapter.

*         Lines 127, 175 and so forth   - for an easier reading, I recommend positioning the reference immediately after the authors' names: Ostendorf et al. [35]

*         Lines 151 and 178 - I recommend placing the figure immediately after the first citation in the text. Follow the same rule for tables.

*         I suggest processing Figure 2 for a better visualization of the anatomical elements indicated by the numbers.

*         The abbreviations should be arranged separately.

Best regards.

Author Response

Referee 1:

Dear authors,

Thank you for the opportunity to review the manuscript entitled “Magnetic Resonance Imaging (MRI)-Based Semi-Quantitative Methods for Rheumatoid Arthritis: From Scoring to Measurement. “

From a structural point of view, the present work falls into the category of narrative review. The manuscript is easy to read, the scientific information being presented in a logical sequence.

All the chapters of the manuscript are exhaustive and provides important data regarding the debated topic. The Introduction is well documented and provides the necessary background to accommodate the reader with the subject of the study. The purpose of the article is presented clearly and concisely. The Materials and Methods must be reviewed and some additional data related to the included items must be entered. As a radiologist, I particularly appreciate the Results/Discussions chapter, which stands out for the way it is presented and for the details provided. I think that this work is useful for both radiologists and rheumatologists. The conclusions are well formulated and reflect the results of the study. 

Thank you for appreciating the manuscript.

In addition, I identified several aspects that can improve the quality of this article:

*         Lines 82-85 - contain the same information as lines 77-80. I recommend merging them in the Introduction chapter.

Thank you, lines 82-85 have been deleted.

*         Lines 127, 175 and so forth   - for an easier reading, I recommend positioning the reference immediately after the authors' names: Ostendorf et al. [35]

The references have been repositione when citing the first author’s name. Thank you.

*         Lines 151 and 178 - I recommend placing the figure immediately after the first citation in the text. Follow the same rule for tables.

Thank you for your comment. However, the layout of figures and tables will be up to the journal's graphic designers.

*         I suggest processing Figure 2 for a better visualization of the anatomical elements indicated by the numbers.

Thank you for the suggestion. Figure 2 has been modified by adding a legend for erosions (asterisks) and synovial hypertrophy (white circles).

*         The abbreviations should be arranged separately.

Thank you for your comment, however, we do not fully understand what the reviewer means. Each abbreviation is given in full in the text when first used.

Best regards.

Reviewer 2 Report

Comments and Suggestions for Authors

In this exposé of existing semiquantitative MRI scoring system, several concerns arise, that need to be addressed:

Page 2, row 59: For consistency reasons (since tenosynovitis is mentioned as a recent addition), JSN should also be mentioned as an addition.

Page 2 row 95: The features of MRI is compared with ultrasound. Although it is true that MRI allows assessment of all joints, the following two points are not true. Ultrasound assessment is standardized according to the OMERACT scoring system (although the assessment is user-dependent). Furthermore, ultrasound is well-suited for quantification of different pathologies, as defined in the scoring systems. Please correct this statement.

Page 3 row 120: Regarding the statement that RAMRIS has potential for low reproducibility among inexperienced readers. Isn’t this true for all available scoring systems and imaging modalities? Can the authors refer to a scoring system that shows high reproducibility among inexperienced readers? If these exist, they should be presented. Otherwise, I suggest that the statement is omitted.

Page 3 row 141: The definition of BME is incorrect in terms of suggested MRI sequence. According to the RAMRIS atlas, BME should be assessed on a T2-weighted fat-saturated sequence, or a STIR sequence if the latter is not available. Please, correct this, and make sure that this error is not reproduced throughout the manuscript.

 Page 4 row 181: This section is rather confusing. It seems like the authors describe the calculation of subscores as a part of the RAMRIS-5 scoring system, while the referred article (reference 19) describes the procedure of calculating RAMRIS subscores to compare correlation between the scoring systems. I suggest that this sentence is deleted.

Page 4 row 198: Here, the authors refer to reference 18, stating that the number of “study areas” are reduced from 116 (RAMRIS) to 36 (SAMIS). It is highly unclear how they reach the given number of study areas in RAMRIS, but I struggle to reach this number by counting myself. I suggest that these numbers are omitted or that the authors define how they reach the number 116.

Page 5 row 209: Again, there is an important difference between T2-weighted images and T2 weighted fat-saturated images. The latter were used in SAMIS. Please correct this.

Page 5 row 242: There seems to be missing a number 2 in the description of how synovial proliferation was scored. Please, correct this.

Figure 1: The symbols for JSN (colored circle) and bone erosion (square) are incorrect and should switch names (JSN is visualized by the square in the figure and bone erosion by the colored circle).

Figure 1: Flexor tendons are incorrectly named. Tendon nr 1 is Flexor carpi radialis, tendon nr 2 is Flexor pollicis longus and flexor compartment 3 is the Ulnar bursa. Tendon number 4 is not depicted in the figure and is also not included in the scoring system. Please correct this.

Figure 3: In the text explaining figure B and C, all bones included in the scoring system are mentioned, except Trapezoideum, which is marked in the figure. Please correct this.

Author Response

Referee 2:

In this exposé of existing semiquantitative MRI scoring system, several concerns arise, that need to be addressed:

 Page 2, row 59: For consistency reasons (since tenosynovitis is mentioned as a recent addition), JSN should also be mentioned as an addition.

Amended, thank you.

Page 2 row 95: The features of MRI is compared with ultrasound. Although it is true that MRI allows assessment of all joints, the following two points are not true. Ultrasound assessment is standardized according to the OMERACT scoring system (although the assessment is user-dependent). Furthermore, ultrasound is well-suited for quantification of different pathologies, as defined in the scoring systems. Please correct this statement.

Amended, thank you.

Page 3 row 120: Regarding the statement that RAMRIS has potential for low reproducibility among inexperienced readers. Isn’t this true for all available scoring systems and imaging modalities? Can the authors refer to a scoring system that shows high reproducibility among inexperienced readers? If these exist, they should be presented. Otherwise, I suggest that the statement is omitted.

Amended, thank you.

Page 3 row 141: The definition of BME is incorrect in terms of suggested MRI sequence. According to the RAMRIS atlas, BME should be assessed on a T2-weighted fat-saturated sequence, or a STIR sequence if the latter is not available. Please, correct this, and make sure that this error is not reproduced throughout the manuscript.

Amended, thank you.

Page 4 row 181: This section is rather confusing. It seems like the authors describe the calculation of subscores as a part of the RAMRIS-5 scoring system, while the referred article (reference 19) describes the procedure of calculating RAMRIS subscores to compare correlation between the scoring systems. I suggest that this sentence is deleted.

Thank you, sentence delete and reworded for the sake of clarity.

Page 4 row 198: Here, the authors refer to reference 18, stating that the number of “study areas” are reduced from 116 (RAMRIS) to 36 (SAMIS). It is highly unclear how they reach the given number of study areas in RAMRIS, but I struggle to reach this number by counting myself. I suggest that these numbers are omitted or that the authors define how they reach the number 116.

Amended, thank you.

Page 5 row 209: Again, there is an important difference between T2-weighted images and T2 weighted fat-saturated images. The latter were used in SAMIS. Please correct this.

Amended, thank you.

Page 5 row 242: There seems to be missing a number 2 in the description of how synovial proliferation was scored. Please, correct this.

Amended, thank you.

Figure 1: The symbols for JSN (colored circle) and bone erosion (square) are incorrect and should switch names (JSN is visualized by the square in the figure and bone erosion by the colored circle).

Thank you for pointing out these errors. Figure 1 has been changed in accordance.

Figure 1: Flexor tendons are incorrectly named. Tendon nr 1 is Flexor carpi radialis, tendon nr 2 is Flexor pollicis longus and flexor compartment 3 is the Ulnar bursa. Tendon number 4 is not depicted in the figure and is also not included in the scoring system. Please correct this.

Thank you for pointing out these errors. Figure 1 legend has been changed in accordance.

Figure 3: In the text explaining figure B and C, all bones included in the scoring system are mentioned, except Trapezoideum, which is marked in the figure. Please correct this.

Thank you. Trapezoidum is now included in the Figure 3B and 3C legends.